# Burnout and its associated factors among healthcare workers in COVID-19 isolation centres in Khartoum, Sudan: A cross-sectional study

Esraa S. A. Alfadul[1]*, Malaz Mohammed Idrees Abdalmotalib[1], Salma Salah Khalid Alrawa[1], Rama Osman Abdelrahman Osman[1], Hadiea Mosaab AhmedElbashir Hassan[1], Alsamany taha albasheir[1], Elfatih A. Hasabo[1], Sagad O. O. Mohamed[1], Kamil Mirghani Ali Shaaban[2]

1 Faculty of Medicine, University of Khartoum, Khartoum, Sudan, 2 Department of Community Medicine, Faculty of Medicine, University of Khartoum, Khartoum, Sudan

* Esraaalfadul.uofk@gmail.com

## Abstract

### Background

Burnout prevalence and its consequences on healthcare workers during the Omicron wave are not well investigated in Sudan. This study aims to assess the prevalence of burnout and its associated factors among doctors and nurses during the omicron wave in COVID-19 isolation centres in Khartoum, Sudan.

### Method

This cross-sectional survey study was conducted at multiple COVID-19 isolation centres in Khartoum state during the omicron wave of Coronavirus Disease 2019 between 20th February 2022 and 10th April 2022. A total of 306 doctors and nurses filled out the questionnaire, with a response rate of 64.8. They were recruited from 5 isolation centers scattered in the three cities of Khartoum Metropolis. The level of burnout was assessed using an online semi-structured questionnaire based on the Oldenburg Burnout Inventory questionnaire. Descriptive statistics were used for continuous variables and frequencies with percentages for categorical variables. The Chi-square test and Fisher exact test were used to identify variables associated with burnout. Logistic regression was used to determine the factors associated with burnout, and the p-value of $\leq .05$ is considered statistically significant.

### Results

The prevalence of burnout was 45.7%. Doctors were more likely to have burnout than nurses (OR: 2.01, CI 95% 1.24–3.27; p = 0.005). Also, married healthcare workers were more likely to suffer burnout than single healthcare workers (OR: 3.89, CI 95% 1.41–12.5; P = 0.013). The number of household members (p = 0.035) was associated with burnout among participants.

**Data Availability Statement:** "All relevant data are within the paper and its Supporting Information files."

**Funding:** The author(s) received no specific funding for this work.

**Competing interests:** The authors have declared that no competing interests exist.

## Conclusion

There is a high prevalence of burnout among healthcare workers in Khartoum Isolation Centers, which is more apparent among doctors.

## Introduction

The Coronavirus Disease 2019 (COVID-19) is an infectious respiratory disease caused by the SARS-CoV-2 virus with varying degrees of morbidity and mortality depending on the age and fitness of the individual [1]. The virus emerged on 31st December 2019 in Wuhan, Hubei Province, China [2] and was declared a global health emergency on 30th January 2020 and a global pandemic on 11th March 2020 by the World Health Organization [3–5] Sudan reported the first COVID-19 case on 13th March 2020; after the declaration of the first case, the Sudan Federal Ministry of Health enhanced the measures to combat the spread of the virus. On 13th April, after the registration of 10 cases of COVID-19, the authorities announced a total lockdown in the state of Khartoum—the most populated state in Sudan -suspending all gatherings, even prayers in mosques, as the virus spread beyond Khartoum, reaching other states. On 25th May, five isolation centers were established in Khartoum to isolate and deal with COVID-19 cases. Later, the number of isolation centers increased in proportion to the high infection rate of the virus, reaching to 14,401 cases with 1,116 deaths by 11th November 2020, reflecting a high fatality rate of 7.7%. By 7th July 2022, there were 62,745 confirmed cases of COVID-19, with 4,952 deaths in Sudan [6–9].

Although public health measures, such as lockdowns, and social distancing, are crucial to reducing the spread of COVID-19, they were found to increase stress, anxiety, and mental disorders [10]. The World Health Organization has recognized "burnout" as an "occupational phenomenon", as lockdowns have significantly affected our work-life balance and work environment. Much research has ascertained burnout and its contributing factors [11].

COVID-19 has a critical psychological impact on the community [12]. The length of the pandemic period -more than two years-, the low evidence regarding the virus treatment protocols, and the unknown destiny of the pandemic resulted in many studies that showed a significant increase in cases of anxiety, psychological stress, and depressive disorders worldwide.

The medical staff worked under severe psychological pressure: being highly stressed by losing patients and colleagues, preference of having long shifts to protect their families, having no clear prevention strategies, and urgency in instructing interpretation, all incredibly highlight peaking of burnout among healthcare providers [13–16].

Burnout is a critical issue that generates inefficiency in healthcare organizations [13] It lowers the quality of healthcare and negatively impacts patient prognosis [17]. Burnout affects the psychological well-being of the staff leading to medical errors. It makes the health system waste a lot of money and resources- because some workers leave their jobs, compelling the system to recruit new staff and offer a new training program. It impedes the process of psychological support for patients, which is part of the treatment [17, 18] Healthcare providers who are emotionally exhausted -express burnout- can neither support patients psychologically nor make fateful medical decisions.

While the pandemic affects the world, studies show that developed countries are affected less than developing ones in terms of fatality, as it has a higher rate of transmission to elderly coupled with poor access to healthcare facilities [19]. Africa is the most vulnerable region for the impact of higher mortality and morbidity, especially among healthcare providers [16].

Sudan's revolution, the terrible economic status, collapsing health system, end-stage arrival of cases due to the centralized isolation centers, difficulty of transportation, and stigmatization of COVID-19 cases all affect the mental well-being of healthcare providers. There is no present data regarding burnout among healthcare providers in Sudan. Many published papers confirmed the high prevalence of burnout -in various countries during the pandemic due to diversified factors- leaving no doubt that the crisis is more exacerbated in Sudan [20, 21].

The term 'Burnout' entails emotional exhaustion, reduced feelings of achievement, and we personalization [22]. According to studies, the prevalence of burnout among healthcare providers varies from 49.3% to 58% worldwide, and intensivists, emergency department doctors, and nurses are particularly susceptible [23].

Burnout prevalence and its consequences on healthcare workers are not well investigated in Sudan. Still, the stigmatization of mental problems in Sudan and the professional stigma of mental illness proposes that healthcare providers repress their mental suffering, navigating to more emotional stress and susceptibility to burnout [24]. Sudan's health system has been facing critical challenges, including a lack of personal protective equipment and other necessary medical equipment. As a result, healthcare workers must deal with COVID-19 without the basic equipment to protect the patients and themselves from the infection. Consequently, this puts tremendous pressure on the doctors, largely due to the poor infection control policies and frustration over being unable to deliver the best possible patient care [25].

The extent of burnout, its distribution between workers in different Isolation centers, and its predictors and protective factors are essential for decision-makers. In our study, we will measure the prevalence of burnout among healthcare providers exposed to COVID-19 patients, so we can contribute with data helping decision-makers fight the battle of the COVID-19 pandemic and the hidden pandemic -burnout -.

## Material and methods

### Study design and settings

This study employed a facility-based cross-sectional design conducted from February 20th, 2022 to April 10th, 2022, during the Omicron wave of the COVID-19 pandemic. The study focused on ten isolation centers situated in Khartoum state, encompassing the cities of Khartoum, Omdurman, and Bahri, which together form the Sudanese Metropolis. The selected isolation centers were Ibrahim Malik Teaching Hospital, Haj El-Mardi Hospital, Al-Shaab Teaching Hospital, Omdurman, Alnaw Hospital, Ahmed Gasim Teaching Hospital, Bahri Teaching Hospital, Jabra Hospital for emergency and injuries, Omdurman, and Khartoum Teaching Hospital. To ensure representation, the largest and smallest isolation centers were chosen from each of the three cities, based on number of beds(capacity).

However, due to Ahmed Gasim Hospital being inactive during the Omicron wave, only two centers, Alnaw Hospital and Omdurman Hospital from Omdurman, Al-Shaab Teaching Hospital, and Jabra Hospital for emergency and injuries from Khartoum, alongside Bahri Teaching Hospital from Bahri, were included in the study.

### Participants

The participants in this study consisted of all doctors and nurses working in the selected centers during the designated period, with no exclusions. Convenience sampling was employed for recruitment, wherein collaborators from each facility were contacted, and online surveys using Google Forms were provided to them for distribution among healthcare practitioners. Two reminders were sent to encourage participation. The targeted number of healthcare practitioners was 472, comprising 182 doctors and 290 nurses. Ultimately, 306 respondents,

including doctors and nurses, completed the questionnaire, resulting in a response rate of 64.8%.

## Data collection and tool

Data were collected using an online self-administered questionnaire distributed by Google Forms. The questionnaire was self-assembled and piloted for clarity and practicality. The questionnaire was structured into five sections. The first section encompassed socio-demographic characteristics, such as age, gender, residence, marital status, number of household members, presence of older family members in the household, history of comorbidities, and history of mental illness. The second section focused on professional characteristics, including the isolation center, job title, years of experience, involvement in ICU, working hours, extra duties, and experience with COVID-19 infection. The third section explored predictors of burnout, including exposure to COVID-19 among relatives, colleagues, the morbid status of the patients. fear of infection, satisfaction with hospital protective measures, and feelings of despair regarding COVID-19 patients. The fourth section examined adaptive behaviors among participants, such as exercise, spirituality, smoking, professional support, self-care, mental breaks, and procrastination. Finally, the fifth section assessed burnout using the Oldenburg Burnout Inventory (OLBI) questionnaire [26], a validated tool comprising 16 items. The items consisted of positively and negatively worded questions related to exhaustion and disengagement, recorded on a four-point Likert scale ranging from 1 (Strongly Agree) to 4 (Strongly Disagree). A higher score indicated a higher level of burnout. Disengagement refers to distancing oneself from the content of one's work, while exhaustion refers to fatigue, loss of energy, feelings of emptiness, and a strong need for rest [27–29]. Participants meeting the thresholds of 2.1 or higher for exhaustion subscales and 2.25 or higher for disengagement subscales, with a cut-off point of 35 [29], were considered at high risk of burnout.

## Ethical approval

Ethical approval for conducting this research was obtained from the Ministry of Health—Khartoum state / Directorate General of Curative Medicine / No: 44, Sudan.

The participants were asked to give consent that they agree to participate in the study by filling the questionnaire for research purposes in the online form, and all the participants provided informed written consent after providing a clear explanation of the study purpose. Data collection was anonymized, and the confidentiality of the study participants was maintained.

## Statistical analysis plan

Data were extracted in an excel sheet, cleaned, and imported into the R software version 4.0.2 and SPSS version 28 (SPSS Inc., Chicago, IL, USA). The normality of distribution was tested using Kolmogorov- Smirnov test. Descriptive statistics were used for calculating the mean and Standard deviation for the continuous variables and frequencies with percentages for categorical variables. The Chi-square test and Fisher exact test were used to identify variables associated with burnout. A multiple logistic regression analysis was performed to identify factors associated with a state of burnout. Those the variables that showed a statistically significant relationship at the bivariate analysis level were included at the multiple analysis level. The p-value of $\leq .05$ was set as the significance level of the study.

## Results

### Participants information

A total of 306 healthcare workers, with the majority being between 26–30 years. More than half of the participants were males (n = 156, 51.0%), and the majority were living in Omdurman (45.1%), followed by Khartoum (42.8%). Also, most of them were singles (83.3%), nurses (58.5%), and only 30 participants (9.8%) were suffering from comorbidities. Regarding the status of COVID-19 in their relatives, 39% stated that their relatives had been infected with COVID-19, while 17.0% were infected and then died. **(Table 1).**

Most respondents were from Jabra Isolation Center (39%) and Omdurman Teaching Hospital (33%). About 54% of the participants reported working in an ICU and 23% reported having COVID-19. **(Table 1).**

### Burnouts among participants

Responses to the Oldenburg Burnout Inventory (OLBI) are shown in **(Table 2).** Burnout prevalence was 45.7% among study participants. The number of household members (p = 0.035) and being a doctor (p = 0.003) were associated with burnout among participants. Also, marital status was significantly associated with burnout (p = 0.001) **(Table 1).** Not surprisingly, fear of infection (p < 0.001) and satisfaction with hospital safety measures (p = 0.002) were associated with burnout.

The most frequent adaptive behaviors were taking brief mental breaks throughout the day (42.2%) and exercising (37.6%) **(Table 3).** Interestingly the following factors were not statistically significantly associated with burnout: average income, working years, working hours, extra hours, previous working experience in COVID-19 centers, working center, working site, and fear of patient death despite all measures. **(Table 1).**

The multiple logistic regression showed doctors were more likely to have burnout than nurses (OR: 2.01, CI 95% 1.24–3.27; p = 0.005). Also, married healthcare workers were more likely to suffer burnout than single healthcare workers (OR: 3.89, CI 95% 1.41–12.5; P = 0.013) **(Table 4).**

## Discussion

Currently, the spread of the novel coronavirus has been deemed a major source of uncertainty, fear and anxiety for a lot of healthcare workers around the world, affecting their physical and psychological health [30]. The current study has assessed burnout among doctors and nurses and provided a better understanding of the problem in sudan. Approximately half of the doctors and nurses in the COVID-19 isolation centers in Khartoum that were included in the present study suffered high levels of work-related stress and burnout in terms of emotional exhaustion and a lack of personal accomplishment and engagement. Physical and emotional well-being of healthcare staff is critical to pandemic containment. However, high burnout rates pose a significant threat to the delivery of safe and effective healthcare, and negatively influence healthcare providers, patients, and the healthcare system.

The prevalence of burnout reported in the current study is In line with previous studies assessing burnout among Sudanese healthcare workers during COVID-19. For instance, 71% of resident physicians in Sudan met the criteria for burnout using the OLBI tool [31]. Another study using the Maslach Burnout Inventory assessment tool revealed that 86.1% of resident physicians in Gezira State's teaching hospitals had burnout syndrome [32]. Further, these findings align with other countries that have reported high rates of burnout among healthcare workers during COVID-19 pandemic [33–37].

**Table 1. Baseline characteristics and demographic data of included participants.**

| Variables | N | Overall, N = 306[1] | burnout No, N = 166[1] | Yes, N = 140[1] | p-value[2] |
|---|---|---|---|---|---|
| **Age** | 306 | | | | 0.5 |
| 20–25 | | 118 (39%) | 65 (39%) | 53 (38%) | |
| 26–30 | | 163 (53%) | 85 (51%) | 78 (56%) | |
| More than 30 | | 25 (8.2%) | 16 (9.6%) | 9 (6.4%) | |
| **Gender** | 306 | | | | 0.8 |
| Female | | 150 (49%) | 80 (48%) | 70 (50%) | |
| Male | | 156 (51%) | 86 (52%) | 70 (50%) | |
| **Residency** | 306 | | | | 0.9 |
| Bahari | | 37 (12%) | 19 (11%) | 18 (13%) | |
| Khartoum | | 131 (43%) | 70 (42%) | 61 (44%) | |
| Omdurman | | 138 (45%) | 77 (46%) | 61 (44%) | |
| **Marital status** | 306 | | | | **0.001** |
| Single | | 255 (83%) | 140 (84%) | 115 (82%) | |
| Widowed | | 3 (1.0%) | 1 (0.6%) | 2 (1.4%) | |
| Engaged | | 26 (8.5%) | 20 (12%) | 6 (4.3%) | |
| Married | | 22 (7.2%) | 5 (3.0%) | 17 (12%) | |
| **Number of household members** | 299 | | | | **0.035** |
| <5 | | 55 (18%) | 21 (13%) | 34 (24%) | |
| >10 | | 19 (6.4%) | 12 (7.5%) | 7 (5.0%) | |
| 5–10 | | 225 (75%) | 127 (79%) | 98 (71%) | |
| **Do you live with an elderly household member who has a chronic disease?** | 306 | | | | 0.3 |
| No | | 175 (57%) | 90 (54%) | 85 (61%) | |
| Yes | | 131 (43%) | 76 (46%) | 55 (39%) | |
| **Do you suffer from any comorbidities?** | 306 | | | | 0.4 |
| No | | 276 (90%) | 152 (92%) | 124 (89%) | |
| Yes | | 30 (9.8%) | 14 (8.4%) | 16 (11%) | |
| **Do you have a history of a mental illness?** | 306 | | | | 0.3 |
| No | | 299 (98%) | 164 (99%) | 135 (96%) | |
| Yes | | 7 (2.3%) | 2 (1.2%) | 5 (3.6%) | |
| **Do you have relative/s who have been infected/died from Covid-19?** | 306 | | | | 0.7 |
| Infected | | 118 (39%) | 64 (39%) | 54 (39%) | |
| Infected and died | | 52 (17%) | 31 (19%) | 21 (15%) | |
| None | | 136 (44%) | 71 (43%) | 65 (46%) | |
| **Do you have colleague/s who have been infected/died from Covid-19?** | 306 | | | | 0.8 |
| Infected | | 230 (75%) | 127 (77%) | 103 (74%) | |
| Infected and died | | 17 (5.6%) | 8 (4.8%) | 9 (6.4%) | |
| None | | 59 (19%) | 31 (19%) | 28 (20%) | |
| **What was your COVID-19 Screening test result?** | 306 | | | | 0.088 |
| Not tested | | 131 (43%) | 72 (43%) | 59 (42%) | |
| Tested negative | | 106 (35%) | 64 (39%) | 42 (30%) | |
| Tested positive | | 69 (23%) | 30 (18%) | 39 (28%) | |
| **What is your Job title?** | 306 | | | | **0.003** |
| Doctor | | 127 (42%) | 56 (34%) | 71 (51%) | |
| Nurse | | 179 (58%) | 110 (66%) | 69 (49%) | |
| **What is your perceived average income?** | 306 | | | | >0.9 |

*(Continued)*

**Table 1.** (Continued)

| Variables | N | Overall, N = 306[1] | burnout | | p-value[2] |
|---|---|---|---|---|---|
| | | | No, N = 166[1] | Yes, N = 140[1] | |
| High | | 5 (1.6%) | 3 (1.8%) | 2 (1.4%) | |
| Low | | 116 (38%) | 64 (39%) | 52 (37%) | |
| Medium | | 185 (60%) | 99 (60%) | 86 (61%) | |
| **How many years have you been working in this job?** | 306 | | | | 0.7 |
| <2 years | | 148 (48%) | 77 (46%) | 71 (51%) | |
| >6 years | | 4 (1.3%) | 2 (1.2%) | 2 (1.4%) | |
| 2–6 years | | 154 (50%) | 87 (52%) | 67 (48%) | |
| **In which isolation center do you work?** | 306 | | | | 0.8 |
| AlNaw Teaching Hospital | | 19 (6.2%) | 12 (7.2%) | 7 (5.0%) | |
| AlShaab Hospital | | 44 (14%) | 21 (13%) | 23 (16%) | |
| Bahri Hospital | | 23 (7.5%) | 12 (7.2%) | 11 (7.9%) | |
| Jebra Isolation center | | 118 (39%) | 65 (39%) | 53 (38%) | |
| Omdurman Teaching Hospital | | 102 (33%) | 56 (34%) | 46 (33%) | |
| **What was your way of employment?** | 306 | | | | >0.9 |
| Appointed by the Ministry of Health | | 49 (16%) | 26 (16%) | 23 (16%) | |
| Attachment | | 22 (7.2%) | 13 (7.8%) | 9 (6.4%) | |
| Compulsory service | | 19 (6.2%) | 11 (6.6%) | 8 (5.7%) | |
| Contracts: appointed by the hospital | | 216 (71%) | 116 (70%) | 100 (71%) | |
| **Did you work in isolation centers in the previous Covid-19 waves other than the current Omicron wave?** | 306 | | | | 0.6 |
| No | | 105 (34%) | 55 (33%) | 50 (36%) | |
| Yes | | 201 (66%) | 111 (67%) | 90 (64%) | |
| **Do you work extra duty or extra hours per week?** | 306 | | | | 0.065 |
| No | | 112 (37%) | 53 (32%) | 59 (42%) | |
| Yes | | 194 (63%) | 113 (68%) | 81 (58%) | |
| **In the isolation center, do you work in the ward or in the intensive care unit (ICU)?** | 306 | | | | 0.6 |
| ICU | | 164 (54%) | 91 (55%) | 73 (52%) | |
| The ward | | 142 (46%) | 75 (45%) | 67 (48%) | |
| **Working hours per week** | 306 | | | | >0.9 |
| 40–60 | | 144 (47%) | 79 (48%) | 65 (46%) | |
| Less than 40 | | 109 (36%) | 58 (35%) | 51 (36%) | |
| More than 60 | | 53 (17%) | 29 (17%) | 24 (17%) | |
| **I'm afraid of getting infected by Covid-19 throughout my work.** | 306 | | | | <0.001 |
| Strongly Agree | | 85 (28%) | 61 (37%) | 24 (17%) | |
| Agree | | 44 (14%) | 29 (17%) | 15 (11%) | |
| Neutral | | 99 (32%) | 44 (27%) | 55 (39%) | |
| Disagree | | 43 (14%) | 19 (11%) | 24 (17%) | |
| Strongly Disagree | | 35 (11%) | 13 (7.8%) | 22 (16%) | |
| **I feel safe with the protective measures taken by the hospital.** | 306 | | | | 0.002 |
| Strongly Agree | | 71 (23%) | 46 (28%) | 25 (18%) | |
| Agree | | 77 (25%) | 44 (27%) | 33 (24%) | |
| Neutral | | 90 (29%) | 48 (29%) | 42 (30%) | |
| Disagree | | 41 (13%) | 23 (14%) | 18 (13%) | |
| Strongly Disagree | | 27 (8.8%) | 5 (3.0%) | 22 (16%) | |
| **I feel the patient would die no matter what I did.** | 306 | | | | >0.9 |

*(Continued)*

**Table 1.** (Continued)

| Variables | N | Overall, N = 306[1] | burnout | | p-value[2] |
|---|---|---|---|---|---|
| | | | No, N = 166[1] | Yes, N = 140[1] | |
| Strongly Agree | | 17 (5.6%) | 10 (6.0%) | 7 (5.0%) | |
| Agree | | 34 (11%) | 19 (11%) | 15 (11%) | |
| Neutral | | 66 (22%) | 38 (23%) | 28 (20%) | |
| Disagree | | 92 (30%) | 49 (30%) | 43 (31%) | |
| Strongly Disagree | | 97 (32%) | 50 (30%) | 47 (34%) | |

[1]n (%)

[2]Pearson's Chi-squared test; Fisher's exact test

The high rates of burnout in this study can be explained by the psychological demands of the profession and the high level of socioeconomic pressure and work-related stress that lead healthcare workers to burnout [35]. The country's weak healthcare system, economic meltdown, security situation, political instability and conflicts during this period have placed more pressure on the healthcare system and professionals [36–38]. Moreover, dealing with a high number of patients with poor infrastructure, lack of suitable accommodation and transportation during the lockdown period, concerns about the lack of treatment supplies and personal protective equipment, and the fear of contracting the disease and infecting their families add to the problem [38–40]. Nevertheless, COVID-19 has been ongoing for over two years, and the constant strain may exhaust healthcare workers' coping mechanisms.

Most of the socio-demographic and other variables assessed in this study were not statistically associated with burnout, reflecting the wide distribution of the problem among the participants despite such differences. We found that a lower number of household members, unmarried individuals, and physician jobs were the only variables associated with burnout among the participants. However, previous studies revealed inconsistent results on burnout risk factors among healthcare workers, and it is unclear whether that burnout follows a specific socio-demographic pattern among healthcare workers [31, 33, 37]. However, some of these associations with socio-demographic factors can be explainable. Among different medical professions, there are differences in the working duties, which put doctors at higher risk of violence and abuse. In our setting, doctors are the front-liners who deal with angry patients and their relatives, particularly when counselling them about their COVID-19 diagnosis and condition or when breaking the news that a patient has passed away [25, 39]. Furthermore, they are responsible for dealing with critically ill patients with a higher risk of complications and mortality [25].

Regarding the habits used to lessen burnout symptoms, seeking professional support to cope with moral distress and grief was significantly associated with lower rates of burnout among the participants. The participants have used other habits to alleviate the burnout symptoms, such as exercising, spiritual habits, and taking brief mental breaks throughout the day. However, none showed a significant association with burnout rates in this study.

The findings of this study acknowledge a high demand for appropriate support interventions and necessitate the search for more effective strategies to be implemented to reduce the risk of burnout among healthcare workers. Using different approaches to reduce burnout, such as training to improve stress-coping skills, can avoid or recover from emotional exhaustion and disengagement. In addition, managing the workplace and environment, improving systems, providing regular psychosocial support, and recruiting additional workers can help address organizational and workplace defects because hostile Working conditions contribute

**Table 2. Responses to questions of the burnout tool.**

| Variables | Overall, N = 306[1] | Burnout | | p-value[2] |
|---|---|---|---|---|
| | | No, N = 166[1] | Yes, N = 140[1] | |
| **I always find new and interesting aspects in my work.** | | | | **<0.001** |
| Strongly Agree | 113 (36.9%) | 76 (45.8%) | 37 (26.4%) | |
| Agree | 118 (38.6%) | 62 (37.3%) | 56 (40.0%) | |
| Disagree | 51 (16.7%) | 18 (10.8%) | 33 (23.6%) | |
| Strongly Disagree | 24 (7.8%) | 10 (6.0%) | 14 (10.0%) | |
| **There are days when I feel tired before I arrive at work.** | | | | **<0.001** |
| Strongly Agree | 135 (44.1%) | 94 (56.6%) | 41 (29.3%) | |
| Agree | 108 (35.3%) | 56 (33.7%) | 52 (37.1%) | |
| Disagree | 42 (13.7%) | 9 (5.4%) | 33 (23.6%) | |
| Strongly Disagree | 21 (6.9%) | 7 (4.2%) | 14 (10.0%) | |
| **It happens more and more often that I talk about my work in a negative way.** | | | | **0.005** |
| Strongly Agree | 62 (20.3%) | 43 (25.9%) | 19 (13.6%) | |
| Agree | 99 (32.4%) | 59 (35.5%) | 40 (28.6%) | |
| Disagree | 77 (25.2%) | 33 (19.9%) | 44 (31.4%) | |
| Strongly Disagree | 68 (22.2%) | 31 (18.7%) | 37 (26.4%) | |
| **After work, I tend to need more time than in the past in order to relax and feel better.** | | | | **<0.001** |
| Strongly Agree | 137 (44.8%) | 98 (59.0%) | 39 (27.9%) | |
| Agree | 92 (30.1%) | 39 (23.5%) | 53 (37.9%) | |
| Disagree | 52 (17.0%) | 23 (13.9%) | 29 (20.7%) | |
| Strongly Disagree | 25 (8.2%) | 6 (3.6%) | 19 (13.6%) | |
| **I can tolerate the pressure of my work very well.** | | | | **<0.001** |
| Strongly Agree | 113 (36.9%) | 77 (46.4%) | 36 (25.7%) | |
| Agree | 121 (39.5%) | 63 (38.0%) | 58 (41.4%) | |
| Disagree | 54 (17.6%) | 21 (12.7%) | 33 (23.6%) | |
| Strongly Disagree | 18 (5.9%) | 5 (3.0%) | 13 (9.3%) | |
| **Lately, I tend to think less at work and do my job almost mechanically.** | | | | **<0.001** |
| Strongly Agree | 71 (23.2%) | 57 (34.3%) | 14 (10.0%) | |
| Agree | 123 (40.2%) | 68 (41.0%) | 55 (39.3%) | |
| Disagree | 66 (21.6%) | 22 (13.3%) | 44 (31.4%) | |
| Strongly Disagree | 46 (15.0%) | 19 (11.4%) | 27 (19.3%) | |
| **I find my work to be a positive challenge.** | | | | **<0.001** |
| Strongly Agree | 131 (42.8%) | 89 (53.6%) | 42 (30.0%) | |
| Agree | 109 (35.6%) | 57 (34.3%) | 52 (37.1%) | |
| Disagree | 48 (15.7%) | 15 (9.0%) | 33 (23.6%) | |
| Strongly Disagree | 18 (5.9%) | 5 (3.0%) | 13 (9.3%) | |
| **During my work, I often feel emotionally drained.** | | | | **<0.001** |
| Strongly Agree | 99 (32.4%) | 72 (43.4%) | 27 (19.3%) | |
| Agree | 104 (34.0%) | 59 (35.5%) | 45 (32.1%) | |
| Disagree | 63 (20.6%) | 18 (10.8%) | 45 (32.1%) | |
| Strongly Disagree | 40 (13.1%) | 17 (10.2%) | 23 (16.4%) | |
| **Over time, one can become disconnected from this type of work.** | | | | **<0.001** |
| Strongly Agree | 93 (30.4%) | 67 (40.4%) | 26 (18.6%) | |
| Agree | 110 (35.9%) | 61 (36.7%) | 49 (35.0%) | |
| Disagree | 70 (22.9%) | 25 (15.1%) | 45 (32.1%) | |
| Strongly Disagree | 33 (10.8%) | 13 (7.8%) | 20 (14.3%) | |
| **After working, I have enough energy for my leisure activities.** | | | | 0.3 |

*(Continued)*

**Table 2.** (Continued)

| Variables | Overall, N = 306[1] | Burnout | | p-value[2] |
|---|---|---|---|---|
| | | No, N = 166[1] | Yes, N = 140[1] | |
| Strongly Agree | 38 (12.4%) | 25 (15.1%) | 13 (9.3%) | |
| Agree | 79 (25.8%) | 44 (26.5%) | 35 (25.0%) | |
| Disagree | 77 (25.2%) | 37 (22.3%) | 40 (28.6%) | |
| Strongly Disagree | 112 (36.6%) | 60 (36.1%) | 52 (37.1%) | |
| **Sometimes I feel sickened by my work tasks.** | | | | **<0.001** |
| Strongly Agree | 83 (27.1%) | 63 (38.0%) | 20 (14.3%) | |
| Agree | 109 (35.6%) | 63 (38.0%) | 46 (32.9%) | |
| Disagree | 81 (26.5%) | 30 (18.1%) | 51 (36.4%) | |
| Strongly Disagree | 33 (10.8%) | 10 (6.0%) | 23 (16.4%) | |
| **After my work, I usually feel worn out and weary.** | | | | **<0.001** |
| Strongly Agree | 133 (43.5%) | 100 (60.2%) | 33 (23.6%) | |
| Agree | 97 (31.7%) | 49 (29.5%) | 48 (34.3%) | |
| Disagree | 56 (18.3%) | 15 (9.0%) | 41 (29.3%) | |
| Strongly Disagree | 20 (6.5%) | 2 (1.2%) | 18 (12.9%) | |
| **This is the only type of work that I can imagine myself doing.** | | | | **<0.001** |
| Strongly Agree | 57 (18.6%) | 45 (27.1%) | 12 (8.6%) | |
| Agree | 67 (21.9%) | 49 (29.5%) | 18 (12.9%) | |
| Disagree | 106 (34.6%) | 48 (28.9%) | 58 (41.4%) | |
| Strongly Disagree | 76 (24.8%) | 24 (14.5%) | 52 (37.1%) | |
| **Usually, I can manage the amount of my work well.** | | | | **<0.001** |
| Strongly Agree | 112 (36.6%) | 75 (45.2%) | 37 (26.4%) | |
| Agree | 140 (45.8%) | 73 (44.0%) | 67 (47.9%) | |
| Disagree | 41 (13.4%) | 15 (9.0%) | 26 (18.6%) | |
| Strongly Disagree | 13 (4.2%) | 3 (1.8%) | 10 (7.1%) | |
| **I feel more and more engaged in my work.** | | | | **<0.001** |
| Strongly Agree | 76 (24.8%) | 63 (38.0%) | 13 (9.3%) | |
| Agree | 140 (45.8%) | 79 (47.6%) | 61 (43.6%) | |
| Disagree | 65 (21.2%) | 18 (10.8%) | 47 (33.6%) | |
| Strongly Disagree | 25 (8.2%) | 6 (3.6%) | 19 (13.6%) | |
| **When I work, I usually feel energized** | | | | **<0.001** |
| Strongly Agree | 79 (25.8%) | 59 (35.5%) | 20 (14.3%) | |
| Agree | 131 (42.8%) | 64 (38.6%) | 67 (47.9%) | |
| Disagree | 67 (21.9%) | 31 (18.7%) | 36 (25.7%) | |
| Strongly Disagree | 29 (9.5%) | 12 (7.2%) | 17 (12.1%) | |
| **Score** | 34.2 ± 5.3 | 30.4 ± 3.1 | 38.8 ± 3.5 | **<0.001** |

[1] n (%)

[2] Pearson's Chi-squared test; Fisher's exact test

Note: 1 = Strongly Agree; 2 = Agree; 3 = Disagree; 4 = Strongly Disagree

to stress and job dissatisfaction, resulting in depersonalization, emotional exhaustion, and lack of accomplishment [40, 41].

## Strengths and limitations

This is one of the few studies assessing the burnout burden among doctors and nurses in Sudan. The sample size of the participants, considered a low response population, is

**Table 3. Adaptive behaviour among the participants.**

| Variables | Overall, N = 306[1] | Burnout | | p-value[2] |
|---|---|---|---|---|
| | | No, N = 166[1] | Yes, N = 140[1] | |
| Exercise | 115 (37.6%) | 61 (36.7%) | 54 (38.6%) | 0.7 |
| spiritual habits | 107 (35.0%) | 63 (38.0%) | 44 (31.4%) | 0.2 |
| Smoking | 35 (11.4%) | 18 (10.8%) | 17 (12.1%) | 0.7 |
| Seek professional support to cope with moral distress and grief | 41 (13.4%) | 32 (19.3%) | 9 (6.4%) | **0.001** |
| Take brief mental breaks throughout the day | 129 (42.2%) | 75 (45.2%) | 54 (38.6%) | 0.2 |
| Performing Self-care Routines | 108 (35.3%) | 56 (33.7%) | 52 (37.1%) | 0.5 |
| Poor Time Management and Procrastination | 80 (26.1%) | 49 (29.5%) | 31 (22.1%) | 0.14 |

[1] n (%)

[2] Pearson's Chi-squared test; Fisher's exact test

representative of generalizability to overall COVID-19 isolation centers in Khartoum, Sudan. The findings of this study need to be considered in the context of some limitations; as this is a cross-sectional study, it will be challenging to draw causative relationships. The self-reported nature of the study could raise the possibility of recall bias. Also, the study was done in five sites which might limit results generalization for all healthcare workers in all settings in the country and could compromise representativeness.

## Conclusion

Our study has shown that nearly half of the healthcare workers in isolation centers in Khartoum have suffered from burnout during COVID-19. Several socio-demographic factors have contributed to the increased level of burnout, and multiple coping mechanisms have accounted for a lower level of burnout among healthcare workers. The findings of this study address the high demand for appropriate interventions to be implemented to reduce the risk of burnout among frontline healthcare workers.

**Table 4. Multiple logistic regression for predictors of burnout among healthcare workers.**

| variables | OR[1] | 95% CI[1] | | p-value |
|---|---|---|---|---|
| | | Lower Limit | Upper Limit | |
| **Marital status** | | | | |
| Single | — | | | |
| Married | 3.89 | 1.41 | 12.5 | **0.013** |
| Widowed | 2.04 | 0.19 | 45.2 | 0.6 |
| Engaged | 0.38 | 0.13 | 0.96 | 0.052 |
| **Number of household members** | | | | |
| <5 | — | | | |
| 5–10 | 0.66 | 0.34 | 1.25 | 0.2 |
| >10 | 0.54 | 0.17 | 1.65 | 0.3 |
| **What is your Job title?** | | | | |
| Nurse | — | | | |
| Doctor | 2.01 | 1.24 | 3.27 | **0.005** |

[1]OR = Odds Ratio, CI[1] = Confidence interval

## Supporting information

**S1 Data. Anonymised dataset.**
(XLSX)

## Acknowledgments

We want to thank Ahmed Abdalhalim Mohamed Omer, Omer Abdulmajed, Aya Zakareya Noor Hamid, Asmaa Mohamed Abbas, Aldoma Mohammed Adam Ismail, Elhadi elsiddig Elhadi, Mohammed AFEFE Hasan Salim, and Tagwa Badawe for their help in collecting the data and Mazin S. Haroun for his help in the research. Also, we would like to thank healthcare workers for participating in our study.

## Author Contributions

**Conceptualization:** Esraa S. A. Alfadul.

**Data curation:** Elfatih A. Hasabo.

**Formal analysis:** Elfatih A. Hasabo.

**Funding acquisition:** Esraa S. A. Alfadul.

**Investigation:** Esraa S. A. Alfadul, Salma Salah Khalid Alrawa.

**Methodology:** Esraa S. A. Alfadul, Malaz Mohammed Idrees Abdalmotalib, Salma Salah Khalid Alrawa, Rama Osman Abdelrahman Osman, Hadiea Mosaab AhmedElbashir Hassan, Alsamany taha albasheir, Sagad O. O. Mohamed.

**Project administration:** Esraa S. A. Alfadul.

**Resources:** Esraa S. A. Alfadul, Elfatih A. Hasabo.

**Software:** Elfatih A. Hasabo.

**Supervision:** Kamil Mirghani Ali Shaaban.

**Visualization:** Elfatih A. Hasabo.

**Writing – original draft:** Esraa S. A. Alfadul, Malaz Mohammed Idrees Abdalmotalib, Salma Salah Khalid Alrawa, Rama Osman Abdelrahman Osman, Hadiea Mosaab AhmedElbashir Hassan, Alsamany taha albasheir, Elfatih A. Hasabo, Sagad O. O. Mohamed.

**Writing – review & editing:** Esraa S. A. Alfadul, Malaz Mohammed Idrees Abdalmotalib, Salma Salah Khalid Alrawa, Rama Osman Abdelrahman Osman, Hadiea Mosaab AhmedElbashir Hassan, Alsamany taha albasheir, Elfatih A. Hasabo, Sagad O. O. Mohamed.

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
