## [Decision Letter · Decision Letter 0]

22 Dec 2022

PONE-D-22-28346Burnout and its associated factors among healthcare workers in Khartoum's COVID-19 isolation centers: A cross-sectional studyPLOS ONE

Dear Dr. Hasabo,

Thank you for submitting your manuscript to PLOS ONE. After careful consideration, we feel that it has merit but does not fully meet PLOS ONE’s publication criteria as it currently stands. Therefore, we invite you to submit a revised version of the manuscript that addresses the points raised during the review process.

We look forward to receiving your revised manuscript.

Kind regards,

Syed Ghulam Sarwar Shah, M.B.B.S., M.A., M.Sc., Ph.D.

Academic Editor

PLOS ONE

Journal Requirements:

Additional Editor Comments:

Please address all of the reviewers' concerns and resubmit your revised manuscript.

Reviewers' comments:

Reviewer's Responses to Questions

**Comments to the Author**

1. Is the manuscript technically sound, and do the data support the conclusions?

Reviewer #1: Yes

Reviewer #2: Yes

2. Has the statistical analysis been performed appropriately and rigorously? 

Reviewer #1: Yes

Reviewer #2: Yes

3. Have the authors made all data underlying the findings in their manuscript fully available?

Reviewer #1: No

Reviewer #2: Yes

4. Is the manuscript presented in an intelligible fashion and written in standard English?

Reviewer #1: No

Reviewer #2: Yes

5. Review Comments to the Author

Reviewer #1: The manuscript although it gets the meaning across is not very well written. It should be re-written to bring the standard close to that of the Journal's. Discussion Section in particular can be improved; although studies have been included but their findings have not been connected with those of the study being discussed.

Reviewer #2: Your research is well planned and the methodology is well described. Research can be replicated and therefore there is nothing to add.

Also, your manuscript is interesting but I need you to answer some minor questions:

-In abstract you should show prevalence of burnout

-There should not be any abbreviations in the abstract.

-In introduction you should give a definition of job burnout

-You should describe more literature on their association and on their prevalence in healthcare professionals during the covid 19 pandemic

-The discussion needs significant reworking. It does not explicate the reasons for nurses to have burnout.

-Many bibliographies are obsolete. The bibliographic citations used are more than 5 years old. The authors must update and arrange the bibliography.

-Some references are incomplete or have errors. The authors should review this section.

6. PLOS authors have the option to publish the peer review history of their article (what does this mean?). If published, this will include your full peer review and any attached files.

Reviewer #1: No

Reviewer #2: No

---

## [Author Response · Author response to Decision Letter 0]

5 Feb 2023

27-janouary-2023

Author’s response to reviews

Subject: Revision and resubmission of PONE-D-22-28346

Dear reviewer,

On behalf of all the contributing authors, I would like to express our sincere appreciation of the reviewers’ constructive comments concerning our article entitled “Burnout and its associated factors among healthcare workers in Khartoum's COVID-19 isolation centers: A cross-sectional study”. We have revised our manuscript according to the reviewers’ comments, questions, and suggestions. We believe that the manuscript has been further improved and the revised paragraphs were highlighted in yellow in the manuscript . If there are any other modifications we could make, we would very much like to modify them and we appreciate your help. Below are point-by-point responses to the reviewers' comments.

Your sincerely

Reviewer #1: The manuscript, although it gets the meaning across, is not very well written. It should be re-written to bring the standard close to that of the Journal's. The Discussion Section in particular can be improved; although studies have been included but their findings have not been connected with those of the study being discussed.

response to reviewer 1:

Thank you for your comment. The manuscript was revised and corrected, and the discussion section was improved according to your note.

Reviewer #2:

Comment: In abstract you should show prevalence of burnout

Response: done

Comment: -There should not be any abbreviations in the abstract.

Response: all abbreviations in the abstract were removed

Comment: -In introduction you should give a definition of job burnout

Response: a brief description of burnout was added (end of page 3)

Comment: -You should describe more literature on their association and on their prevalence in healthcare professionals during the covid 19 pandemic.

Response: more description was added to the introduction section (end of page 3)

Comment: -The discussion needs significant reworking. It does not explicate the reasons for nurses to have burnout.

Response: discussion section was revised and edited according to your note. We think the reviewer meant ‘doctors’ not nurses in his comment (see table 4)

Comment: Many bibliographies are obsolete. The bibliographic citations used are more than 5 years old. The authors must update and arrange the bibliography. Some references are incomplete or have errors. The authors should review this section

Response: we re-checked all references to ensure that all of them are not older than 2018

---

## [Decision Letter · Decision Letter 1]

11 Apr 2023

PONE-D-22-28346R1Burnout and its associated factors among healthcare workers in Khartoum's COVID-19 isolation centers: A cross-sectional studyPLOS ONE

Dear Dr. Alfadul,

Thank you for submitting your manuscript to PLOS ONE. After careful consideration, we feel that it has merit but does not fully meet PLOS ONE’s publication criteria as it currently stands. Therefore, we invite you to submit a revised version of the manuscript that addresses the points raised during the review process.

We look forward to receiving your revised manuscript.

Kind regards,

Syed Ghulam Sarwar Shah, M.B.B.S., M.A., M.Sc., Ph.D.

Academic Editor

PLOS ONE

Additional Editor Comments (if provided):

Please address the following issues:

Title: The authors need to revise the title especially because of ‘Khartoum's COVID-19 isolation centers; which could be changed to ‘COVID-19 isolation centres in Khartoum, Sudan’.

ABSTRACT

Background: Please change ‘Khartoum isolation centers’ to ‘COVID-19 isolation centers in Khartoum. Were these centres in Khartoum City or Khartoum state? Please report as appropriate.

Methods: Please what do you mean by ‘multi Centre facility-based? Please revise the following sentence: ‘A multi Centre facility-based cross-sectional study was conducted between 20th February and 10th April 2022 during the last wave of Coronavirus Disease 2019.’ You might like to say: This cross-sectional survey study was conducted at multiple COVID-19 isolation centres in Khartoum during the omicron wave of Coronavirus Disease 2019 between 20th February 2022 and 10th April 2022.

Methods: Please report who is included in ‘healthcare workers’ in your study.

Methods: You mention the ‘last wave’ in the methods while in the background you state the ‘omicron wave’, please use the term consistently.

Methods: Please report the sample type and total sample size (how many participants were invited to the survey) and then report the number.

Methods: Please report the number of respondents and the response rate in the results section. Hence, move the following info to the results section. “A total of 306 doctors and nurses filled out the questionnaire, with a response rate of 64.8℅.”

Methods: Please report the main parts of the survey and how it was administered.

Results: Please change ‘Multivariate logistic regression showed doctors’ to ‘Doctors were …’

Results: What do you mean by engaged healthcare workers? Are these married or not? Please revise the following sentence: Also, engaged healthcare workers were less likely to suffer from burnout than single healthcare workers.

Results: Please what do you mean by ‘A low number of household members’? Could you state the exact number of household members?

Results: You report in the conclusion that there was a high prevalence of burnout in healthcare workers but there is no such information in the results section. Could you please report the prevalence of burnout in your sample.

Conclusion: Please remove the following sentence because it is unclear and does not make a sense.

INTRODUCTION

1. The authors write that the ‘The Coronavirus Disease 2019 (COVID-19) is a respiratory disease’. Could you please report what type of a respiratory disease it is and what is it’s causative agent.

2. Grammar: The authors have used the present tense in the first paragraph while they has used the past tense in the subsequent paras. Please use the same tense, preferably the past tense because you are reporting the past activities/actions.

3. Language: Could you please change the term ‘attack rate’ to ‘infection rate’ or some other appropriate term in the following sentence: “…the high attack rate of the virus….”

4. Language: please change ‘reaching 14,401 cases..’ to ‘reaching to 14,401 cases….’

5. Language: Could you please make a connection between the following sentences:

The World Health Organization has recognized "burnout" as an "occupational phenomenon. And "As lockdowns have significantly affected our work-life balance and work environments, much research has ascertained burnout and its contributing factors.1research has ascertained burnout and its contributing factors.11

6. Language: What do you mean by “the shortage of data about the virus treatment”? Please revise.

7. Long sentences: Please avoid using very long sentence such as the following sentence, which should be divided in 2-3 small sentences:

“ COVID-19 has a critical psychological impact on the community.12 The length of the pandemic period more than two years -, the shortage of data about the virus treatment, and the unknown destiny of the pandemic resulted in many studies that showed a significant increase in cases of anxiety, psychological stress, and depressive disorders worldwide, and because the medical staff worked under severe psychological pressure: being highly stressed by losing patients and colleagues, preference of having long shifts to protect their families, having no clear curing strategies, and urgency in instructing interpretation, all incredibly highlight peaking of burnout among healthcare providers.13-16.”

8. Language: Please what do you mean by ‘It lowers the quality of healthcare systems’? Do you mean ‘It lowers the quality of healthcare? If so, please revise the sentence.

9. Change: Please change ‘cannot support patients psychologically nor make fateful medical decisions’ to ‘can neither support patients psychologically nor make fateful medical decisions’.

10. Clarification: The authors state that “While the pandemic affects the world, studies show that developed countries are affected less than developing ones.” Could you please add more information i.e. in which ways the pandemic has affected developing countries more than the developed countries.

11. 1Revise: Could you please revise the following sentence, perferably by dividing it in 2 sentences: “Although, at present, no data considering the burnout among healthcare providers in Sudan, many published papers confirmed the high prevalence of burnout -among various countries during the pandemic due to diversified factors - leaving no doubt that the crisis is more exacerbated in Sudan.”

12. Check: Please which studies are you referring to in the following sentence: ‘These studies attributed the risk of burnout among healthcare providers to work-related, pandemic-related, and socio-demographic factors. “ This sentence does not seem to be relevant here. If so, please remove it.

METHODS

1. The authors state that “This facility-based cross-sectional study” but they do not state what type of study it is.

2. Please check the dates in the following sentence: ‘between 20th February 2022 and 10th April 2020’.

3. Could the authors describe what they mean by Sudanese metropolis because have used different terms like Khartoum state, Khartoum and Sudanese metropolis.

4 Please refer to ‘from the three parts of the Sudanese metropolis’ What do you mean by the ‘parts’? Do you mean districts or subdistricts of Khartoum city or Khartoum state?

5. Could you please report the criteria for determining the largest and the smallest isolation centers.

6. Could you please report the names of three parts of the Sudanese metropolis/Khartoum that are included in this study.

7. The authors have used different terms such as healthcare workers and healthcare practitioners to describe doctors and nurses. These terms could include other health professionals. Could the authors use doctors and nurse instead of either healthcare professionals or healthcare practitioners. Please be consistent throughout the paper.

8. Could you please report your sampling methods and type.

9. Could you please revise “The burnout level of the participants was assessed using an online self-administered questionnaire” to “Data were collected using an online self-administered questionnaire” because the questionnaire did not collect data on only burnout level but also other variables.

10. Could you please check what do you mean ‘the patient himself’ as a predictor of burnout included in the third part of the questionnaire

11. Please report: how did you recruit and invite the participants? how you administered the survey questionnaire? Did you send any reminders? If yes, how many and when?

!2. Please report whether you developed the survey questionnaire or adapted it from an existing survey. Did you pilot test it before the main study? What was found in the pilot, and did you make any changes in the survey? Who and how many participants were involved in the pilot testing?

13. The author report that the OLBI questionnaire consists of positively

and negatively worded questions. Which questions are negatively worded and how did you manage scores of these negatively worded questions?

14. Could you please check what do you mean by ‘one point designated the lowest burnout and four designated the highest’?

15. Could you please report which online tools did you use for online survey?

16. could you please change: ‘Descriptive statistics were used in mean and Standard deviation…’ to ‘ Descriptive statistics were used for calculating the mean and Standard deviation…’.

17. Could you please report what for did you use the Multiple logistic regression analysis?

RESULTS

1 Please change ‘Among all participants, 22.5% tested positive for COVID-19, and nearly (53.6%) were working in the ICU” to ‘About 54% of the participants reported working in an ICU and 22.5% reported having COVID-19.’

2. The authors report that “Also, marital status was significantly associated with burnout (p = 0.001). Could you please report which marital status was associated with burnout?

3. Could you please explain what do you mean by mental breaks?

4. Please check ‘working hours (0.8) and correct it.

5. Please take out p values information from the following sentence and report that these factors were not statistically significantly associated with burnout and refer to the relevant table.

“Interestingly the following factors were not associated with burnout: average income (p >0.9), working years (p = 0.8), working hours (0.8), extra hours (p = 0.065), previous working experience in COVID-19 centers (p = 0.6) working center ( p = 0.8) and site (p= 0.6) and fear of patient death despite all measure ( p > 0.9)’

6. Could you please describe what do you mean by ‘engaged’ in the following sentence: “Also, engaged healthcare workers were less likely to suffer

burnout than unmarried healthcare workers”. Does it mean engaged but not married yet. Please revise it appropriately.

7. Refer to table 4, which shows that married participants had higher ORs for burnout compared to respondents who were single. Could you high light this in the results in the abstract and main results section.

7. In Table 1, please report categories of the following variables and re-do statistical analysis.7

- Age

-Number of household members

- Do you live with an elderly household member who has a chronic disease?

- Do you suffer from any comorbidities?

- Do you have a history of mental illness?

8. Table 1: Please check your sub-groups for years of working in this job. The categories are overlapping each other, which should not. Moreover, what is 0 years? Please re-categories this variable as < 2 years, 2-6 years and more than 6 years. Then please rerun the statistical tests to check whether there are any statistically significant differences.

9. Table 1. In the following Qs, you have reported the number of participants who reported yes only. Please report the data about the participants who answered No to these Qs.

- Did you work in isolation centers in the previous Covid-19 waves other than the current Omicron wave?

- Do you work extra duty or extra hours per week?

10. Table 1: Please report what are the answer options for the following Q: Working hours per week? Then rerun the data analysis in 2-3 major categories.

11. Table 1. Could you please report a range of local currency for high, medium and low income.

12. Table 1 and 2. Could you please report agree, disagree etc instead of numbers 1, 2 …5

13. Table 3: This table includes different things but the title says habits. Habits for what? Could you please revise the title that shows items/variables reported in this table.

14. Table 4. Please add categories of number of family members and re-run the analysis.

DISCUSSION

1. The first two sentences report almost the same information so could you please revise them. Also provide some references to support these statements.

2. The authors state that ‘The current study addressed a significant issue’ but do not report the name of the issue. Could you please say that you have studied burnout.

3. please change ‘a low-income country’ to ‘Sudan’.

4. please change ‘health staff’ to ‘healthcare staff’

5. please change ‘can be explainable’ to ‘can be explained’.

6. please check whether the following statement is in relation to Sudan or in general because there are several studies on burnout in Drs and nurses during the COVID-19 pandemic. Therefore, please correct the statement. “This is one of the few studies assessing the burnout burden among healthcare workers.”

REFERENCES

Please report abbreviated names of the following journals:

-International Journal of Environmental Research and Public Health

- Cochrane Database of Systematic Reviews

- Archives of Rehabilitation Research and Clinical Translation

Reviewers' comments:

Reviewer's Responses to Questions

**Comments to the Author**

1. If the authors have adequately addressed your comments raised in a previous round of review and you feel that this manuscript is now acceptable for publication, you may indicate that here to bypass the “Comments to the Author” section, enter your conflict of interest statement in the “Confidential to Editor” section, and submit your "Accept" recommendation.

Reviewer #1: All comments have been addressed

Reviewer #2: All comments have been addressed

2. Is the manuscript technically sound, and do the data support the conclusions?

Reviewer #1: Yes

Reviewer #2: Yes

3. Has the statistical analysis been performed appropriately and rigorously? 

Reviewer #1: Yes

Reviewer #2: Yes

4. Have the authors made all data underlying the findings in their manuscript fully available?

Reviewer #1: Yes

Reviewer #2: Yes

5. Is the manuscript presented in an intelligible fashion and written in standard English?

Reviewer #1: Yes

Reviewer #2: No

6. Review Comments to the Author

Reviewer #1: (No Response)

Reviewer #2: Thank you for your great work.

7. PLOS authors have the option to publish the peer review history of their article (what does this mean?). If published, this will include your full peer review and any attached files.

Reviewer #1: No

Reviewer #2: **Yes: **Rasoul Goli

---

## [Author Response · Author response to Decision Letter 1]

27 Jun 2023

24 June 2023

Author’s response to reviews

Subject: Revision and resubmission of PONE-D-22-28346

Dear Editor,

On behalf of all the contributing authors, I would like to express our sincere appreciation of the reviewers’ constructive comments concerning our article entitled “Burnout and its associated factors among healthcare workers in COVID-19 isolation centres in Khartoum, Sudan: A cross-sectional study”. We have revised our manuscript according to the reviewers’ comments, questions, and suggestions. We believe that the manuscript has been further improved and the revised paragraphs were highlighted in yellow in the manuscript . If there are any other modifications we could make, we would very much like to modify them and we appreciate your help. Below are point-by-point responses to the reviewers' comments.

Your sincerely

Title: The authors need to revise the title especially because of ‘Khartoum's COVID-19 isolation centers; which could be changed to ‘COVID-19 isolation centres in Khartoum, Sudan’.

We thank the editor for recommending a better sentence to clarify the meaning, we used the recommended sentence. 

ABSTRACT

Background: Please change ‘Khartoum isolation centers’ to ‘COVID-19 isolation centers in Khartoum. Were these centres in Khartoum City or Khartoum state? Please report as appropriate.

Done

Methods: Please what do you mean by ‘multi Centre facility-based? 

Please revise the following sentence: ‘A multi Centre facility-based cross-sectional study was conducted between 20th February and 10th April 2022 during the last wave of Coronavirus Disease 2019.’ You might like to say: This cross-sectional survey study was conducted at multiple COVID-19 isolation centres in Khartoum during the omicron wave of Coronavirus Disease 2019 between 20th February 2022 and 10th April 2022.

We thank the editor for recommending a better sentence to clarify the meaning, we used the recommended sentence in the abstract and made use of it in the method section.

Methods: Please report who is included in ‘healthcare workers’ in your study.

The included healthcare workers were doctors and nurses , after your note we updated the term.

Methods: You mention the ‘last wave’ in the methods while in the background you state the ‘omicron wave’, please use the term consistently.

We thank the editor for this suggestion. We adopted the (omicron wave) term throughout the study.

Methods: Please report the sample type and total sample size (how many participants were invited to the survey) and then report the number.

Methods: Please report the number of respondents and the response rate in the results section. Hence, move the following info to the results section. “A total of 306 doctors and nurses filled out the questionnaire, with a response rate of 64.8℅.”

The sampling technique and response rate are updated in the methods section.

Methods: Please report the main parts of the survey and how it was administered.

The questionnaire has five sections and are detailed in (data collection and tool) section of the method.

Results: Please change ‘Multivariate logistic regression showed doctors’ to ‘Doctors were …’

Done

Results: What do you mean by engaged healthcare workers? Are these married or not? Please revise the following sentence: Also, engaged healthcare workers were less likely to suffer from burnout than single healthcare workers.

We define engaged healthcare providers as individuals who are in a committed relationship but not married. However, we found that this distinction does not significantly impact the results compared to married healthcare workers, who have a higher odds ratio than their single counterparts. Therefore, we have removed the above sentence and replaced it with the following: "Married healthcare workers were more likely to suffer burnout compared to single healthcare workers (OR: 3.89, CI 95% 1.41–12.5; P = 0.013)."

Results: Please what do you mean by ‘A low number of household members’? Could you state the exact number of household members?

We updated the sentence to be "The number of household members (p= 0.035) was associated with burnout among participants." please refer to table 1 

Results: You report in the conclusion that there was a high prevalence of burnout in healthcare workers but there is no such information in the results section. Could you please report the prevalence of burnout in your sample.

The prevalence of burnout was reported in results section and highlighted

Conclusion: Please remove the following sentence because it is unclear and does not make sense.

Do you mean this sentence "Significant relations show a positive effect on burnout".

If it is, we removed it. 

INTRODUCTION:

1. The authors write that the ‘The Coronavirus Disease 2019 (COVID-19) is a respiratory disease’. Could you please report what type of a respiratory disease it is and what is it’s causative agent.

We updated the sentence to be: Coronavirus Disease 2019 (COVID-19) is an infectious respiratory disease caused by the SARS-CoV-2 virus.

2. Grammar: The authors have used the present tense in the first paragraph while they has used the past tense in the subsequent paras. Please use the same tense, preferably the past tense because you are reporting the past activities/actions.

We changed the present tense in the first paragraph to past tense and highlighted the verbs.

3. Language: Could you please change the term ‘attack rate’ to ‘infection rate’ or some other appropriate term in the following sentence: “…the high attack rate of the virus….”

We changed it to ‘infection rate’

4. Language: please change ‘reaching 14,401 cases..’ to ‘reaching to 14,401 cases….’

We changed it to ‘reaching to’

5. Language: Could you please make a connection between the following sentences:

The World Health Organization has recognized "burnout" as an "occupational phenomenon. And "As lockdowns have significantly affected our work-life balance and work environments, much research has ascertained burnout and its contributing factors.

The 2 sentences has been connected as follows: ‘The World Health Organization has recognized "burnout" as an "occupational phenomenon" as lockdowns have significantly affected our work-life balance and work environment. Much research has ascertained burnout and its contributing factors.11’

6. Language: What do you mean by “the shortage of data about the virus treatment”? Please revise.

We changed it to ‘the low evidence regarding treatment protocols’.

7. Long sentences: Please avoid using very long sentence such as the following sentence, which should be divided in 2-3 small sentences:

“COVID-19 has a critical psychological impact on the community.12 The length of the pandemic period more than two years -, the shortage of data about the virus treatment, and the unknown destiny of the pandemic resulted in many studies that showed a significant increase in cases of anxiety, psychological stress, and depressive disorders worldwide, and because the medical staff worked under severe psychological pressure: being highly stressed by losing patients and colleagues, preference of having long shifts to protect their families, having no clear curing strategies, and urgency in instructing interpretation, all incredibly highlight peaking of burnout among healthcare providers.13-16.”

We divided the sentence into 3 short sentences as follows: ‘COVID-19 has a critical psychological impact on the community. 12 The length of the pandemic period -more than two years-, the lack of evidence regarding the virus treatment, and the unknown destiny of the pandemic resulted in many studies that showed a significant increase in cases of anxiety, psychological stress, and depressive disorders worldwide. The medical staff worked under severe psychological pressure: being highly stressed by losing patients and colleagues, preference of having long shifts to protect their families, having no clear prevention strategies, and urgency in instructing interpretation, all incredibly highlight peaking of burnout among healthcare providers.13-16’

8. Language: Please what do you mean by ‘It lowers the quality of healthcare systems’? Do you mean ‘It lowers the quality of healthcare? If so, please revise the sentence.

Yes, we meant that ‘it lowers the quality of healthcare’.

9. Change: Please change ‘cannot support patients psychologically nor make fateful medical decisions’ to ‘can neither support patients psychologically nor make fateful medical decisions’.

We changed it from ‘cannot support patients psychologically nor make fateful medical decisions’ to ‘can neither support patients psychologically nor make fateful medical decisions’.

10. Clarification: The authors state that “While the pandemic affects the world, studies show that developed countries are affected less than developing ones.” Could you please add more information i.e. in which ways the pandemic has affected developing countries more than the developed countries.

We updated the sentence to be : "in terms of fatality, as it has a higher rate of transmission to elderly coupled with poor access to healthcare facilities" (16).

11. Revise: Could you please revise the following sentence, preferably by dividing it in 2 sentences: “Although, at present, no data considering the burnout among healthcare providers in Sudan, many published papers confirmed the high prevalence of burnout -among various countries during the pandemic due to diversified factors - leaving no doubt that the crisis is more exacerbated in Sudan.”

We revised the sentence and divided it into 2 sentences as follows: ‘There is no present data regarding burnout among healthcare providers in Sudan. Many published papers confirmed the high prevalence of burnout -among various countries during the pandemic due to diversified factors- leaving no doubt that the crisis is more exacerbated in Sudan.’

12. Check: Please which studies are you referring to in the following sentence: ‘These studies attributed the risk of burnout among healthcare providers to work-related, pandemic-related, and socio-demographic factors. “ This sentence does not seem to be relevant here. If so, please remove it.

We removed this sentence since it is irrelevant.

METHODS

1. The authors state that “This facility-based cross-sectional study” but they do not state what type of study it is.

We update it as recommended in another comment to (This study employed a facility-based cross-sectional design)

2. Please check the dates in the following sentence: ‘between 20th February 2022 and 10th April 2020’.

The dates are correct. As doctors and nurses are busy, the collaborators -who are doctors themselves- asked for a longer period to collect the data.

3. Could the authors describe what they mean by Sudanese metropolis because they have used different terms like Khartoum state, Khartoum and Sudanese metropolis.

We changed it to Sudanese Metropolis across the study, and explained what it means. 

4 Please refer to ‘from the three parts of the Sudanese metropolis’ What do you mean by the ‘parts’? Do you mean districts or subdistricts of Khartoum city or Khartoum state?

We mean cities and have updated the term in the methods

5. Could you please report the criteria for determining the largest and the smallest isolation centers.

This was based on the number of beds (capacity) and updated in the method.

6. Could you please report the names of three parts of the Sudanese metropolis/Khartoum that are included in this study.

We named them and added the update in the method section.

7. The authors have used different terms such as healthcare workers and healthcare practitioners to describe doctors and nurses. These terms could include other health professionals. Could the authors use doctors and nurses instead of either healthcare professionals or healthcare practitioners. Please be consistent throughout the paper.

We used doctors and nurses consistently in this version.

8. Could you please report your sampling methods and type.

We reported the sampling method and type.

9. Could you please revise “The burnout level of the participants was assessed using an online self-administered questionnaire” to “Data were collected using an online self-administered questionnaire” because the questionnaire did not collect data on only burnout level but also other variables.

We thank the reviewer for this suggestion. We updated the sentence.

10. Could you please check what do you mean ‘the patient himself’ as a predictor of burnout included in the third part of the questionnaire 

We meant the morbid status of the patient and updated it 

11. Please report: how did you recruit and invite the participants? how you administered the survey questionnaire? Did you send any reminders? If yes, how many and when?

We updated the recruitment method in the participants section.

12. Please report whether you developed the survey questionnaire or adapted it from an existing survey. Did you pilot test it before the main study? What was found in the pilot, and did you make any changes in the survey? Who and how many participants were involved in the pilot testing?

We adapted it from previous studies. OLBI is a well known used tool that is already validated. We only checked for the clarity and practicality of the questionnaire after adding other sections we updated this in the method.

13. The author reports that the OLBI questionnaire consists of positively and negatively worded questions. Which questions are negatively worded and how did you manage scores of these negatively worded questions? 

The items 2, 3, 4, 6, 8, 9, 11 and 12, are negative so the scale is reversed, with Strongly Agree answers scoring 4 and Strongly Disagree answers scoring 1, then summing up with the scores of others questions and using 35 as cut off to categorize participants as having burnout or not. 

14. Could you please check what do you mean by ‘one point designated the lowest burnout and four designated the highest’?

We removed this sentence since it is really irrelevant. The higher the score, the greater the level of burnout. 

15. Could you please report which online tools did you use for online survey?

We used Google form and updated the method.

16. could you please change: ‘Descriptive statistics were used in mean and Standard deviation…’ to ‘ Descriptive statistics were used for calculating the mean and Standard deviation…’.

We updated the sentence. 

17. Could you please report what for did you use the Multiple logistic regression analysis?

We reported it in the statistical analysis plan 

RESULTS

1 Please change ‘Among all participants, 22.5% tested positive for COVID-19, and nearly (53.6%) were working in the ICU” to ‘About 54% of the participants reported working in an ICU and 22.5% reported having COVID-19.’

Done

2. The authors report that “Also, marital status was significantly associated with burnout (p = 0.001). Could you please report which marital status was associated with burnout?

The association was assessed using a test of significance (Table 1), however "married" status was significantly associated with burnout ((OR: 3.89, CI 95% 1.41 – 12.5; p = 0.013)).(Table 4)

3. Could you please explain what do you mean by mental breaks?

By "mental breaks," we refer to the practice of regularly taking breaks during one's shifts in order to minimize stress. It is not necessary to be a spiritual break. 

4. Please check ‘working hours (0.8) and correct it.

Done

5. Please take out p values information from the following sentence and report that these factors were not statistically significantly associated with burnout and refer to the relevant table.

“Interestingly the following factors were not associated with burnout: average income (p >0.9), working years (p = 0.8), working hours (0.8), extra hours (p = 0.065), previous working experience in COVID-19 centers (p = 0.6) working center ( p = 0.8) and site (p= 0.6) and fear of patient death despite all measure ( p > 0.9)

Done

6. Could you please describe what do you mean by ‘engaged’ in the following sentence: “Also, engaged healthcare workers were less likely to suffer

burnout than unmarried healthcare workers”. Does it mean engaged but not married yet. Please revise it appropriately.

We define engaged healthcare providers as individuals who are in a committed relationship but not married. However, we found that this distinction does not significantly impact the results compared to married healthcare workers, who have a higher odds ratio than their single counterparts. Therefore, we have removed the above sentence and replaced it with the following: "Married healthcare workers were more likely to suffer burnout compared to single healthcare workers (OR: 3.89, CI 95% 1.41–12.5; P = 0.013)."

7. Refer to table 4, which shows that married participants had higher ORs for burnout compared to respondents who were single. Could you highlight this in the results in the abstract and main results section.

Done

7. In Table 1, please report categories of the following variables and re-do statistical analysis.7

- Age

-Number of household members

- Do you live with an elderly member who has a chronic disease?

- Do you suffer from any comorbidities?

- Do you have a history of mental illness?

We reported them and reran the analysis. Please see table (1)

8. Table 1: Please check your sub-groups for years of working in this job. The categories are overlapping each other, which should not. Moreover, what is 0 years? Please re-categories this variable as < 2 years, 2-6 years and more than 6 years. Then please rerun the statistical tests to check whether there are any statistically significant differences.

We used the recommended categories. Please, see table (1)

9. Table 1. In the following Qs, you have reported the number of participants who reported yes only. Please report the data about the participants who answered No to these Qs.

- Did you in isolation centers in the previous Covid-19 waves other than the current Omicron wave?

- Do you work extra duty or extra hours per week 

Done

10. Table 1: Please report what are the answer options for the following Q: Working hours per week? Then rerun the data analysis in 2-3 major categories.

Done

11. Table 1. Could you please report a range of local currency for high, medium and low income.

Due to the absence of standardized income categorization in Sudanese research, each paper tends to utilize its own categorization method. We refrain from using specific numerical values and instead allow participants to categorize their income based on their own perspective. Our belief is that stress is linked to individuals' perception of their income, rather than the actual numerical figures.

12. Table 1 and 2. Could you please report agree, disagree etc instead of numbers 1, 2 …5

Done

13. Table 3: This table includes different things but the title says habits. Habits for what? Could you please revise the title that shows items/variables reported in this table.

Done we changed the title into" Adaptive behavior among the participants"

14. Table 4. Please add categories of number of family members and re-run the analysis.

Done

DISCUSSION

1. The first two sentences report almost the same information so could you please revise them. Also provide some references to support these statements.

We thank the reviewers for this comment, we updated the sentences to be:" Currently, the spread of the novel coronavirus has been deemed a major source of uncertainty, fear and anxiety for a lot of healthcare workers around the world, affecting their physical and psychological health" 

2. The authors state that ‘The current study addressed a significant issue’ but do not report the name of the issue. Could you please say that you have studied burnout.

3. please change ‘a low-income country’ to ‘Sudan’.

We followed your advice in these two comments and updated the sentence to be : "The current study has assessed burnout among doctors and nurses and provided a better understanding of the problem in sudan" 

4. please change ‘health staff’ to ‘healthcare staff’

We updated it to healthcare staff 

5. please change ‘can be explainable’ to ‘can be explained’.

We are sorry about this mistake , it is corrected and highlighted in the discussion

6. please check whether the following statement is in relation to Sudan or in general because there are several studies on burnout in Drs and nurses during the COVID-19 pandemic. Therefore, please correct the statement. “This is one of the few studies assessing the burnout burden among healthcare workers.

We clarified the statement and hence it became : This is one of the few studies assessing the burnout burden among doctors and nurses in Sudan.

REFERENCES

Please report abbreviated names of the following journals:

-International Journal of Environmental Research and Public Health

- Cochrane Database of Systematic Reviews

- Archives of Rehabilitation Research and Clinical Translation 

Done

---

## [Editor Report · Decision Letter 2]

2 Jul 2023

Burnout and its associated factors among healthcare workers in COVID-19 isolation centres in Khartoum, Sudan: A cross-sectional study

PONE-D-22-28346R2

Dear Dr. Alfadul,

We’re pleased to inform you that your manuscript has been judged scientifically suitable for publication and will be formally accepted for publication once it meets all outstanding technical requirements.

Kind regards,

Syed Ghulam Sarwar Shah, M.B.B.S., M.A., M.Sc., Ph.D.

Academic Editor

PLOS ONE

Additional Editor Comments (optional):

Many thanks for submitting the revised manuscript.
---

## [Editor Report · Acceptance letter]

13 Jul 2023

PONE-D-22-28346R2 

Burnout and its associated factors among healthcare workers in COVID-19 isolation centres in Khartoum, Sudan: A cross-sectional study 

Dear Dr. Alfadul:

I'm pleased to inform you that your manuscript has been deemed suitable for publication in PLOS ONE. Congratulations! Your manuscript is now with our production department. 

Kind regards, 

on behalf of

Dr. Syed Ghulam Sarwar Shah 

Academic Editor

PLOS ONE